# Non-Noble-Metal Mono and Bimetallic Composites for Efficient Electrocatalysis of Phosphine Oxide and Acetylene C-H/P-H Coupling under Mild Conditions

**DOI:** 10.3390/ijms24010765

**Published:** 2023-01-01

**Authors:** Maxim V. Tarasov, Olga D. Bochkova, Tatyana V. Gryaznova, Asiya R. Mustafina, Yulia H. Budnikova

**Affiliations:** Arbuzov Institute of Organic and Physical Chemistry, FRC Kazan Scientific Center of RAS, Arbuzov Str. 8, 420088 Kazan, Russia

**Keywords:** electrochemistry, composite, mono and bimetallic catalyst, acetylene, phosphonation

## Abstract

The present work describes an efficient reaction of electrochemical phosphorylation of phenylacetylene controlled by the composition of catalytic nanoparticles based on non-noble-metals. The sought-after products are produced via the simple synthetic protocol based on room temperature, atom-economical reactions, and silica nanoparticles (SNs) loaded by one or two d-metal ions as nanocatalysts. The redox and catalytic properties of SNs can be tuned with a range of parameters, such as compositions of the bimetallic systems, their preparation method, and morphology. Monometallic SNs give phosphorylated acetylene with retention of the triple bond, and bimetallic SNs give a bis-phosphorylation product. This is the first example of acetylene and phosphine oxide C-H/P-H coupling with a regenerable and recyclable catalyst.

## 1. Introduction

In recent years, the development of new efficient catalytic systems based on inexpensive non-noble metals has been an important component of green chemistry [1,2,3]. Bioinspired tandem catalysis based on bimetallic structures has opened a new dimension of efficient catalytic systems, surpassing traditional catalytic methods of organic synthesis, often reducing the number of stages, reaction time and facilitating process conditions [4,5,6,7,8,9]. In addition, the growing demand for environmentally friendly and economical synthetic procedures is also driving the development of multiple catalytic one-pot conversions in order to obtain the desired product in the most efficient manner. In this regard, mono- and bimetallic heterogeneous, nanoheterogeneous catalysts are excellent candidates for selective transformations, with the possibility of regeneration and easy separation of the catalyst from the reaction mixture [10,11].

In bimetallic catalyst systems, two different metals can catalyze two or more different types of reactions or steps, with some synergy between the two metals. Thus, the use of bimetallic catalyst systems can result in an overall increase in reactivity and selectivity over their monometallic counterparts. Stability and adaptability appear to be important factors in the synthesis of these bimetallic catalysts, for which the nature of the support matrix or ligand may play a decisive role. In electrochemical processes, such nanoheterogeneous catalytic structures are used as sensors [12,13], in electrosynthesis of practically significant complex molecules [14,15,16,17,18,19,20,21,22], for activation of CO_2_, N_2_, and other small molecules [23,24].

Substituted alkynes and alkenes are fundamental structural motifs that are widely distributed in natural products, bioactive molecules, and functional materials. They serve as universal synthetic precursors and/or intermediates in organic transformations [25,26,27,28,29,30,31,32,33,34,35,36,37,38]. Despite the long history of the functionalization of alkynes, the search for new ways and catalysts is very relevant. For example, a new strategy has been proposed for multi-metal-catalyzed (Cu, Ni, Ag) oxidative radical alkynylation with terminal alkynes as Sonogashira-type alkynylation for C(sp^3^)−C(sp) bond formation at 80–100 °C, however, the regeneration of the catalytic components is impossible in this case [26].

Phosphorylation of acetylenes is an important way to obtain practically significant ligands, flame retardants, biologically active compounds, for example, the progesterone receptor antagonist [27]. Depending on the conditions and reagents, the products may contain both a carbon-carbon triple bond [28,29,30,31,32,33,34,35,36,37,38,39,40,41,42] and a double [43,44,45,46] or single [47,48,49] bonds upon hydrophosphination. The classical routes to phosphorus-substituted alkynes are mostly based on elimination reactions from the corresponding vinylichalide/pseudohalide derivatives or the reaction of a metal acetylide with a halophosphine or a derivative [33] (Figure 1). Despite the success of their synthesis, most reactions are carried out under harsh and hazardous conditions, techniques include elevated temperatures [36,38,39], excess oxidizing agents and expensive radical initiators (such as DBU [28,32]), catalyst metals, primarily expensive palladium, rhodium and silver [34,35,39,40,42,44], the latter is in excess in many cases, are often based on sulfonyl [29], halogen [32,33,37] and other derivatives [32,35,36,38,41] (Figure 1).

Recoverable and recyclable catalytic systems for these transformations are not described.

In this regard, the development of an efficient and environmentally benign method for the direct construction of C(sp, sp^2^, sp^3^)-P bonds through C-H/P-H coupling of acetylenes and phosphine oxides under noble metal- and oxidant-free conditions is still highly desirable.

The rationale for choosing copper and cobalt as metal components of catalysts is associated with their availability and activity in various phosphorylation reactions, primarily through unsaturated bonds. In particular, the works [30,31,46,48,50,51,52] exemplify the copper- and cobalt-catalyzed reactions correspondingly.

Silica nanoparticles (SNs) were widely applied as convenient nanobeads for different metal ions and complexes. Moreover, metal ions or complexes can be incorporated into SNs through either (1) localization within silica confinement or (2) surface adsorption. These two ways of incorporation allow tuning and exposure of the doped metal ions to a bulk of solution. The present work introduces a synthetic procedure to incorporate cobalt, copper, and iron ions into SNs, thus, producing original mono- and bimetallic nanomaterial for the use as nanocatalyst of the reductive coupling of phenylacetylene and diphenylphosphine oxide through the C-H/P-H reaction. It is worth noting that SNs serve as carriers of metal ions, thus, preventing their leaching and further undesirable transformations. The backgrounds of the bimetallic nanoparticles are the previously developed synthetic procedures allowing the encapsulation of both Co^II^ ions and Co^III^ complexes into the silica nanoparticles (SNs) [13,20]. It will be shown that the catalytically active form of the nanocatalyst is generated electrochemically at the cathode, and the nature (mono or bimetallic) of the catalyst is of great impact on the products of the reaction.

## 2. Results and Discussion

### 2.1. Synthesis of Nanocomposite Catalytic Systems

The composite SNs were synthesized in the framework of core-shell morphology, where metal ions were incorporated within silica spheres through either doping or adsorption procedures. The localization of metal salts or complexes within silica confinement can be realized through their addition into the synthetic mixture, while the incorporation through surface adsorption can be performed under the post-treating of the synthesized SNs. Both microemulsion water-in-oil and Stober procedures are commonly applied techniques for the synthesis of SNs. It is worth noting that the use of either microemulsion or Stober techniques allows controlling both the size and porosity of the silica spheres [53]. The addition of metal complexes or metal ions into the synthetic mixture results in their encapsulation into silica spheres, thus, resulting in the composite SNs. It is worth noting that the Co^III^ ions are incorporated in the form of kinetically inert [Co(dipy)_3_]^3+^ complexes. Thus, their inner-sphere environment remains unchanged after the synthetic procedure [20], resulting in the composite SNs designated as Co^III^@SN_50_ and Co^III^@SN_120_, where 50 and 120 is their size in nanometers. The applied synthetic procedures are described in detail in the Exp. Section and schematically represented in Figure 1. The Co^III^-content is greater in Co^III^@SN_50_ vs. Co^III^@SN_120_, which is the agreement with the previously reported tendency [20]. The doping of Co^II^ ions into the SN_50_ was performed through the microemulsion synthetic procedure [13].

As it has been previously demonstrated, the high activity of silanol groups is the reason for their complexation with d-metal ions, which provides a main driving force of the specific adsorption of d-metal ions [54,55]. The stirring of the empty SN_50_ and SN_120_ in the aqueous solutions of Cu^II^ and Fe^III^ chlorides results in significant adsorption, which is evident from the Si:Cu(Fe) molar ratios in the SNs after their separation from the aqueous solutions and washing (Table 1).

The composites Co^III^@SN_50_ and Co^III^@SN_120_ also demonstrate similar adsorption of both metal ions (Table 1). The metal ions incorporated through the adsorption technique are manifested by the bands in the diffuse reflectance spectra (Figure 2) at the wavelengths (800–900 nm and 400–550 nm) peculiar for the d-d transition of Cu^II^ and Fe^III^, respectively [56].

The DLS measurements of the aqueous colloids of the mono and bimetallic SNs are characterized by negative electrokinetic potential values (Table 1), which argues for the fact that the adsorbed metal ions exhibit rather deep penetration into the silica matrix (Figure 1). The values of the hydrodynamic radius and polydispersity index of the SNs represented in Appendix A indicate the similarity in the aggregation behavior of the bimetallic and monometallic SNs.

### 2.2. Electrocatalytic Phosphorylation of Phenylacetylene

The redox properties of the nanoparticles were studied using cyclic voltammetry (CV) on modified glassy carbon electrodes. Peaks of the corresponding metals are observed on the CVs (Table 2). The redox transitions of Cu^II/I^, Fe^III/II^, Co^III/II^bpy_n_ are usually reversible or quasi-reversible, which indicates the stabilization of the reduced forms of metals in a specific environment. The proximity of the first potentials of the reduction peaks of the metals included in the bimetallic nanoparticles leads to a complex shape of the peaks that cannot be resolved and accurately assigned (Appendix A). Peak potentials are shown in Table 2.

The studied nanoparticles were tested for catalytic activity in the phenylacetylene phosphorylation reaction of phenylacetylene with diphenylphosphine oxide under electroreductive conditions. The Ph_2_P(O)H conversion was 100% in all cases. 

Joint electrolysis of diphenylphosphine oxide with phenylacetylene (1:1) under electrochemical reduction conditions in the presence of the nanocatalyst at room temperature with a background electrolyte Et_4_NBF_4_ in the galvanostatic mode with the passage of 2F electricity proceeds with the formation of diphenyl(phenylethynyl)phosphine oxide (1) in the form of a mixture isomers and (1-phenylethane-1,2-diyl)bis(diphenylphosphine oxide) (2) (Figure 2, Table 2). The cathode potential in all cases was −1.5–1.6 V when passing 2 F of electricity and increased to −1.9 V with further electrolysis. Monitoring of the process by ^31^P NMR spectra showed that after 1F electricity, product (1) and the residual amount of the precursor Ph_2_P(O)H are present in the solution. After the passage of 2 F electricity, product (2) precipitated from the reaction mixture, and the ^31^P NMR spectrum of the solution contained only the signal of compound (**1**) and no traces of the starting Ph_2_P(O)H.

The nature of the catalyst, as it turned out, is decisive for obtaining a particular product. Monometallic catalyst particles favor the formation of phosphorylated acetylene with triple bond (1) (the yield up to 98%, Table 2, entry 1), while the bimetallic catalyst favors the formation of bisphosphorylated addition adduct with saturated carbon-carbon bonds (2) (Figure 3, Appendix A, entry 1, yield of 95%). Increasing the content of Ph_2_P(O)H to 2:1 with respect to phenylacetylene and increasing the amount of electricity passed to 3.5 F makes it possible to obtain a single product (2) with the bimetallic nanocatalysts participation. In the latter case, the reaction is completely atom-economical since there are no formal by-products.

The poor solubility of product **2** in acetonitrile leads to its precipitation, which prevents the easy separation of catalyst nanoparticles and their reuse after electrosynthesis of **2**. However, in the absence of product **2** the electrolyte solution is transparent, the catalyst is easily separated by centrifugation and can be reused. The contamination of catalyst particles with product **2** can be prevented by heating the mixture to 80 °C or by washing the nanoparticles with chloroform. It has been established that an isolated and regenerated catalyst works without loss of activity at least three times.

Bimetallic catalysts promote the hydrogenation of intermediates, which leads to the final conversion of acetylene into product 2 with saturated C–C bonds. Differences in the activity of mono- and bimetallic catalysts have also been observed previously in numerous works [7,57,58], although the reasons for synergy or fundamentally different selectivity and performance are usually difficult to explain. However, in many cases, the bimetallic catalysts promote reactions accompanied by the hydrogenation of a wide variety of molecules and intermediates, as it is exemplified for iron−cobalt [59,60] or cobalt-copper particles [24,61,62,63,64,65].

It was found that the addition of phenylacetylene, which is electrochemically inactive in the potential working region, to catalyst nanoparticles has a significant effect on voltammograms nanoparticles. Thus, for example, new catalytic peaks appear on the CVs of Co^II^–SN_50_–Cu^II^ reduction in the presence of phenylacetylene at lower potentials (Appendix A), and a catalytic increase in current is observed when acetylene is added to the SN_50_-Cu^II^ (Appendix A) and SN_120_–Cu^II^ (Appendix A) particles. The addition of Ph_2_P(O)H, which itself is reduced at high potentials (Ep = −2.89 V, Appendix A), does not noticeably affect the currents and potentials of the first peaks of catalyst reduction. 

The product yields represented in Table 2 indicate the difference in the content of products obtained via the Cu-catalyzed phosphorylation reaction under the use of SN_120_–Cu^II^ and SN_50_–Cu^II^ as nanocatalysts. The specificity of the surfaces of SNs produced via the microemulsion procedure due to the porosity arising from the washing out of the adsorbed TX-100 molecules 53] differentiates SN_50_–Cu^II^ from SN_120_–Cu^II^. The more porous surface of SN_50_–Cu^II^ vs. SN_120_–Cu^II^ can explain the difference in the yields of products **1** and **2** under their use as nanocatalysts. Thus, nanoparticles SN_50_ provide an optimal basis for both mono and bimetallic nanocatalysts.

Product **2** is easily separated from the reaction mixture since it is poorly soluble in it, and it is sufficient to leave the reaction mixture for some time after the reaction to get almost quantitative precipitation of **2**.

(1-phenylethane-1,2-diyl)bis(diphenylphosphine oxide) (**2**) (CAS Number: 3583-85-5) has practical significance and is produced on an industrial scale by several concerns, for example, BaiFuChem, Xiamen Equation Chemical Co., Ltd. и Uhnshanghai (China) (http://www.equationchemical.com (accessed on 31 December 2022), https://www.baifuchem.com (accessed on 31 December 2022), http://www.uhnshanghai.com/uhn/html/20149252096.html (accessed on 31 December 2022)) as farma intermediate or flame retardant. Thus, we propose a simple and convenient method for the synthesis of this product. Furthermore, this article discussed the different activity and catalytic performance of bimetallic SNs compared to monometallic composites. The synthetic strategies reported here established development of sophisticated and controlled SNs for widespread application.

## 3. Conclusions

A method is proposed for the preparation of phosphorylation products of terminal acetylene using the example of phenylacetylene and Ph_2_P(O)H using a regenerated nanocatalyst based on silica nanoparticles loaded by Cu^II^, Co^III/II^, Fe^III^ via different synthetic techniques in both mono- and bimetallic modes. The synthetic technique has been optimized for controlled cross-coupling of phenylacetylene with retention of the triple bond. Moreover, the use of the bimetallic nanocatalysts allows producing bis-adduct with two phosphine oxide substituents and a fully saturated carbon backbone along with the C-H/P-H cross-coupling of phenylacetylene with retention of the triple bond. Both products are in demand and practically significant. They are obtained in one stage at room temperature by an atom-economical reaction.

## Data Availability

All data generated or analyzed during this study are included in this published article; further inquiries can be directed to the corresponding author.

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
