# Peer review of "Non-Noble-Metal Mono and Bimetallic Composites for Efficient Electrocatalysis of Phosphine Oxide and Acetylene C-H/P-H Coupling under Mild Conditions"

_ijms, 2023, doi:10.3390/ijms24010765_

Round 1

Reviewer 1 Report

The manuscript  describes electrochemical phosphorylation of  phenylacetylene catalyzed by metal-containing  nanocomposites. Interestingly, chemoselectivity of the reaction depends on the type of the composite: monometallic particles produce phosphorylated acetylene whereas bimetallic composites give bis-phosphorylated alkane.  Both products are practically important, therefore, both reaction routes are of synthetic value. The reactions occur in mild conditions and require no precious metals for catalysis. Since the direct methods of the C-P bond formation are of great demand, the manuscript is worth publishing. The material is well presented; however, some critical issues remain to be solved.

1.       The yields of the products are given in Table 2 for the experiments when a charge of 2F/mol  was passed. As it is mentioned in page 5, an increase in the electricity passed (3.5 F) and the doubled amount of Ph2P(O)H allow selective obtaining of product 2. The reaction yields should be provided in Scheme 2 or in the main text .

2.       It is unclear from Scheme 2, what bimetallic composites were tested? Are there the same as given in Table 2? What about their relative efficiency?

3.       The reactions were performed in a galvanostatic  regime. And what about potential values? Did you control the potential value during the reaction? This information is important and may shed light on the reaction mechanism.

4.       As it is stated in page 6, “… the addition of phenylacetylene, which is electrochemically inactive in the working potential region, to catalyst nanoparticles has a significant effect on voltammograms in the case of bimetallic nanoparticles and practically does not change the picture for monometallic ones”. However, metal coordination to acetylene takes place in both reaction routes (Scheme 3), regardless of which product is formed. Please, comment on this.

5.       The plausible reaction route given in Scheme 3 does not seem very convincing. In fact, no real support has been provided. Abstraction of H-radicals which are further involved in the other reaction seems doubtful. I can recommend not to provide this scheme and to focus on the synthetic value of the elaborated process (and that is beyond doubt!).

6.       Some misprints  should be corrected (e.g., p.4, line 13; p.6, line 18 (new peaks appear in the CV), etc.)

Author Response

Dear Editors and Reviewers,

We send you the revised manuscript ID: ijms-2141261, Title: «Non-Noble-Metal Mono and Bimetallic Composites for Efficient Electrocatalysis of Phosphine Oxide and Acetylene C-H/P-H Coupling under Mild Conditions»

We would like to thank the Reviewers for fruitful revision of the manuscript. We considered all the remarks and the list of changes that we made in the manuscript is presented below:

Reviewer: 1
1.       The yields of the products are given in Table 2 for the experiments when a charge of 2F/mol was passed. As it is mentioned in page 5, an increase in the electricity passed (3.5 F) and the doubled amount of Ph2P(O)H allow selective obtaining of product 2. The reaction yields should be provided in Scheme 2 or in the main text.

Reply: Corrected. The reaction yields are indicated in the text. Also we have given references to Tables with details: Monometallic catalyst particles favor the formation of phosphorylated acetylene with triple bond (1) (the yield up to 98%, Table 2, entry 1), while the bimetallic catalyst favors the formation of bisphosphorylated addition adduct with saturated carbon-carbon bonds (2) (Scheme 3, Table S3, entry 1, yield of 95%).

  1. It is unclear from Scheme 2, what bimetallic composites were tested? Are there the same as given in Table 2? What about their relative efficiency?

Reply: Yes, the same bimetallic (or monometallic) composites are used in all experiments, indicated in Tables 2 and S3, summarizing the yields of specific products in depending on the nature of the particles and the amount of electricity and the ratio of reagents. We specified in the caption to the Scheme: «Comparison of coupling products with mono- and bimetallic catalytic systems (from Tables 2 and S3).»

  1. The reactions were performed in a galvanostatic  regime. And what about potential values? Did you control the potential value during the reaction? This information is important and may shed light on the reaction mechanism.

Reply: The cathode potential in all cases was -1.5-1.6 V when passing 2 F of electricity, and increased to 1.9 V with further electrolysis. We have added this sentence to the text of the manuscript and Supporting Information.

  1. As it is stated in page 6, “… the addition of phenylacetylene, which is electrochemically inactive in the working potential region, to catalyst nanoparticles has a significant effect on voltammograms in the case of bimetallic nanoparticles and practically does not change the picture for monometallic ones”. However, metal coordination to acetylene takes place in both reaction routes (Scheme 3), regardless of which product is formed. Please, comment on this.

Reply: Thank you very much for this comment. This is a mistake, an inaccurate description of observations. We rewrote this phrase and concretized the details indicating the Figures of voltammograms from SI : «It was found that the addition of phenylacetylene, which is electrochemically inactive in the working potential region, to catalyst nanoparticles has a significant effect on voltammograms nanoparticles. Thus, for example, new catalytic peaks appear on the CVs of CoII – SN50 – CuII reduction in the presence of phenylacetylene at lower potentials (Fig. S9) and a catalytic increase in current is observed when acetylene is added to the SN50-CuII (Fig. S10) and SN120–CuII (Fig.S11) monometallic particles».

  1. The plausible reaction route given in Scheme 3 does not seem very convincing. In fact, no real support has been provided. Abstraction of H-radicals which are further involved in the other reaction seems doubtful. I can recommend not to provide this scheme and to focus on the synthetic value of the elaborated process (and that is beyond doubt!).

Reply: Agree. We have removed the Scheme of the proposed mechanism. We have some evidence in its favor, but they require further clarification and research, which we will publish in the future.

  1. Some misprints  should be corrected (e.g., p.4, line 13; p.6, line 18 (new peaks appear inthe CV), etc.)

Reply: Corrected. We checked the text in detail and tried to correct all typos and inaccuracies.

Reviewer 2 Report

The manuscript discusses using non-noble metal-based catalytic nanoparticles for the electrochemical phosphorylation of phenylacetylene. According to the manuscript, the required compounds may be made using a straightforward synthetic strategy that uses atom-economic reactions at room temperature and silica nanoparticles (SNs) loaded with one or two d-metal ions as nano catalysts. The manuscript mentions that altering variables such the bimetallic systems' composition, the technique of synthesis, and the shape might change the redox and catalytic capabilities of SNs. The manuscript also states that whereas bimetallic SNs can provide a bis-phosphorylation product, monometallic SNs can generate phosphorylated acetylene with preservation of the triple bond. This is the first instance of acetylene and phosphine, according to the manuscript.

There are some issues with the manuscript that requires modifications:

[1] First paragraph is too long it must become divided into two.

[2] The manuscript notes that stability and flexibility are crucial aspects in the production of these catalysts, and that bimetallic catalytic complexes can lead to an overall improvement in reactivity and selectivity over their monometallic counterparts. Provide stats here.

[3] Despite the success of these reactions, the manuscript points out that the majority of phosphorylation methods involve harsh and dangerous conditions, including high temperatures, an abundance of oxidising agents, expensive radical initiators, and the frequent use of catalysts made of pricey metals like palladium. Provide stats on harsh and dangerous condition that comes from other methods.

[4] Font in Figure 1 needs to become bigger.

Author Response

Dear Editors and Reviewers,

We send you the revised manuscript ID: ijms-2141261, Title: «Non-Noble-Metal Mono and Bimetallic Composites for Efficient Electrocatalysis of Phosphine Oxide and Acetylene C-H/P-H Coupling under Mild Conditions»

We would like to thank the Reviewers for fruitful revision of the manuscript. We considered all the remarks and the list of changes that we made in the manuscript is presented below:

Reviewer: 2.

[1] First paragraph is too long it must become divided into two.

Reply: Done.

[2] The manuscript notes that stability and flexibility are crucial aspects in the production of these catalysts, and that bimetallic catalytic complexes can lead to an overall improvement in reactivity and selectivity over their monometallic counterparts. Provide stats here.

Reply: The stats on impact of mono- vs bimetallic catalysts can be found in the reviews which has been cited in the revised version of the MS. However, it is worth noting that the accurate comparative analysis of the catalytic activity of mono- vs bimetallic nanoparticles can be made in the framework of the similar nanoparticulate morphology, since the oxidation state of metal ions, their inner- and outer-sphere environment, as well as the features (size, shape and nature) of polymeric beads are the factors influencing the catalytic activity of mono- and bimetallic nanocatalysts. For example, the widely applied mono- and bimetallic nanocatalysts based on metal oxides cannot be compared with the nanostructures represented in the present work. Thus, the revealed tendency is specific for the core-shell silica nanoparticulate framework. We have clarified and provided relevant references (p.1): «Thus, the use of bimetallic catalyst systems can result in an overall increase in reactivity and selectivity over their monometallic counterparts [9]. Stability and adaptability appear to be important factors in the synthesis of these bimetallic catalysts, for which the nature of the support matrix or ligand may play a decisive role [9-11].»

[3] Despite the success of these reactions, the manuscript points out that the majority of phosphorylation methods involve harsh and dangerous conditions, including high temperatures, an abundance of oxidising agents, expensive radical initiators, and the frequent use of catalysts made of pricey metals like palladium. Provide stats on harsh and dangerous condition that comes from other methods.

Reply: Done. We have done a more detailed state-of-the-art of known methods of phosphorylation, and have focused on the phosphorylation of acetylenes. Thus, a new scheme 1 appeared. Also, clarifications and more detailed references were made in the text:

«Phosphorylation of acetylenes is an important way to obtain practically significant ligands, flame retardants, biologically active compounds, for example, the progesterone receptor antagonist [27]. Depending on the conditions and reagents, the products may contain both a carbon-carbon triple bond [28-42], and a double [43-46] or single [47-49] bonds upon hydrophosphination. The classical routes to phosphorus-substituted alkynes are mostly based on elimination reactions from the corresponding vinylichalide/pseudohalide derivatives or the reaction of a metal acetylide with a halophosphine or a derivative [33] (Scheme 1). Despite the success of their synthesis, most reactions are carried out under harsh and hazardous conditions, techniques include elevated temperatures [36, 38, 39], excess oxidizing agents and expensive radical initiators (such as DBU [28, 32]), catalyst metals, primarily expensive palladium, rhodium and silver [34,35,39,40,42,44], the latter is in excess in many cases, are often based on sulfonyl [29], halogen [32,33,37 ] and others derivatives [32, 35,36, 38, 41] (Scheme 1).»

[4] Font in Figure 1 needs to become bigger.

Reply: Done.

The necessary corrections were indicated in yellow or red color in manuscript.

Finally, with all corrections we have made us truly hope our contribution would match the highest standards of IJMS.

We fully appreciate your help and assistance in correcting the manuscript.